# A Preliminary Prototype High-Speed Feedback Control of an Artificial Cochlear Sensory Epithelium Mimicking Function of Outer Hair Cells

**DOI:** 10.3390/mi11070644

**Published:** 2020-06-29

**Authors:** Hiroki Yamazaki, Dan Yamanaka, Satoyuki Kawano

**Affiliations:** Graduate School of Engineering Science, Osaka University, Toyonaka, Osaka 560-8531, Japan; yamazaki@bnf.me.es.osaka-u.ac.jp (H.Y.); yamanaka@bnf.me.es.osaka-u.ac.jp (D.Y.)

**Keywords:** sensorineural hearing loss, cochlear implant, piezoelectric material, fully implantable device, feedback control, outer hair cell

## Abstract

A novel feedback control technique for the local oscillation amplitude in an artificial cochlear sensory epithelium that mimics the functions of the outer hair cells in the cochlea is successfully developed and can be implemented with a control time on the order of hundreds of milliseconds. The prototype artificial cochlear sensory epithelium was improved from that developed in our previous study to enable the instantaneous determination of the local resonance position based on the electrical output from a bimorph piezoelectric membrane. The device contains local patterned electrodes deposited with micro electro mechanical system (MEMS) technology that is used to detect the electrical output and oscillate the device by applying local electrical stimuli. The main feature of the present feedback control system is the principle that the resonance position is recognized by simultaneously measuring the local electrical outputs of all of the electrodes and comparing their magnitudes, which drastically reduces the feedback control time. In this way, it takes 0.8 s to control the local oscillation of the device, representing the speed of control with the order of one hundred times relative to that in the previous study using the mechanical automatic stage to scan the oscillation amplitude at each electrode. Furthermore, the intrinsic difficulties in the experiment such as the electrical measurement against the electromagnetic noise, adhesion of materials, and fatigue failure mechanism of the oscillation system are also shown and discussed in detail based on the many scientific aspects. The basic knowledge of the MEMS fabrication and the experimental measurement would provide useful suggestions for future research. The proposed preliminary prototype high-speed feedback control can aid in the future development of fully implantable cochlear implants with a wider dynamic range.

## 1. Introduction

The hearing process has been investigated in great detail and its mechanics captured in a basic model develped by Békésy. Even today, many researchers are devoting efforts to understanding the mechanics of audition, and remarkable progress has been made since Békésy’s model, as summarized in Refs. [1,2,3,4]. The human ear consists of the outer, middle, and inner ears. A sound is collected by the outer ear, amplified or dampened by the middle ear [5], and transmitted to the inner ear. The inner ear contains a sensory organ with a spiral shape called the cochlea, which processes the sound. The information contained in the sound consists of its pitch and loudness, as described simply by p=Asin(2πft), where *A* is the amplitude of the sound wave, corresponding to the loudness; *f* is the frequency, corresponding to the pitch; and *t* is time. Additionally, *p* is the acoustic pressure, which is the local deviation of the pressure from the ambient value and represents the mechanical means by which sound travels through media and can be measured by sensors. The frequency is processed by the basilar membrane, which has a trapezoidal shape when the cochlea is unrolled and straightened. The basilar membrane distinguishes the frequencies contained in a sound, acting as an analog Fourier transform analyzer. As a result of its trapezoidal shape and spatially nonhomogeneous mechanical oscillation characteristics, it can separate frequencies based on the spatial position of the resonance peak along the membrane. Two types of hair cells that respond to the oscillation of the basilar membrane in a different ways are arranged along the basilar membrane. The inner hair cells at the resonance position send a neurotransmitter to the spiral ganglion cells, thereby transmitting the acoustic information to the brain through the auditory nerves [6]. The amount of neurotransmitter that is released varies depending on the magnitude of the membrane oscillation. Therefore, the inner hair cells process the loudness and perform the function of converting the mechanical vibration into electrical signals. The outer hair cells control the oscillation of the basilar membrane actively varying their length to amplify small oscillations, giving the basilar membrane a dynamic range of 0–120 dB and improving the frequency selectivity in the range of 20 Hz to 20 kHz [7,8,9]. With its spiral shape and a small volume of approximately 0.19 mL [10,11], the cochlea responds mechanically and electrically to sound vibrations with high accuracy; therefore, it can be said that the cochlea is an exquisite micro electro mechanical systems (MEMS).

A malfunction of the hair cells can cause the inability to detect sound, followed by sensorineural hearing loss, which is one type of deafness and is often recognized as sudden hearing loss in young people or as congenital deafness in new borns. As a result that damaged hair cells never regenerate in mammals [12], cochlear implants are thought to be the best clinical solution for sensorineural hearing loss and are widely commercialized. Trends in cochlear implant technology are outlined in Refs. [13,14], therefore here only a brief description of the cochlear implant is presented. In general, cochlear implants consist of extracorporeal and internal devices. The extracorporeal units collect the sound and sends the processed sound information via the outer coil. Based on the transmitted information, the inner coil indirectly stimulates the auditory nerve via electrodes immersed in the cochlea. Although the cochlear implants has seen design and performance improvements [15,16,17], there are some inconveniences with the extracorporeal apparatus, such as the need for batteries. The external units must be removed when patients are sleeping or the batteries need to be charged. However, for children and infants with these implants if they are not always perceiving sound, their language acquisition ability may deteriorate [18]. Therefore, there is a high demand for fully implantable cochlear implants without extracorporeal devices, and many researchers have attempted to fabricate them using MEMS techniques for future clinical applications in parallel with other technologies such as induced pluripotent stem cells (iPSCs) [19,20,21].

The functions required for fully implantable cochlear implants are frequency selectivity and power generation without batteries. It is key that the device has a trapezoidal shape mimicking the basilar membrane and displays piezoelectric properties. It has been shown that a trapezoidal membrane [22,23,24,25,26,27,28,29] or an array of beams with various lengths [30,31,32,33,34,35] can realize frequency selectivity over a wide range of local resonance frequencies. Piezoelectric materials generate electrical signals in response to an external force making them applicable to fully implantable cochlear implants. Shintaku et al. [36] have shown that poly(vinylidene fluoride–trifluoroethylene) (P(VDF-TrFE)) films are biocompatible and have low toxicity for the human body. In addition, polyvinylidene difluoride (PVDF) [22,23,25,29,33,35], lead zirconate titanate (PZT) [24,34], zinc oxide [26], and aluminum nitride [30,31,32] have also been applied in previous studies. Our and a few research groups have experimentally observed electrically evoked auditory brainstem responses (eABRs) with fully implantable artificial cochleae to evaluate their potential for clinical application [37,38,39]. However, amplifiers producing a 1000-fold [37], 100-fold [38], and 950-fold [39] signal amplification were needed to stimulate the auditory nerves in these studies because the eABR threshold requires an electrical potential on the order of volts but the electrical signals from organic piezoelectric materials are generally small. Some papers have reported the measurement of electrical signals from piezoelectric devices implanted in animal cochleae [37,40]. Maximum voltage outputs of 29.3 μV [37] and 79.7 μV [40] were recorded, and these are small to stimulate the auditory nerves.

Although the effective electrodes that can directly stimulate the spiral ganglion cells have been fabricated and these have decreased the threshold to half of its initial value [41], the feasibility of the resulting piezoelectric device is not satisfactory, and biomechanical methods are necessary to enhance the dynamic range of the device. To overcome the difficulties and unsolved problems with existing implant designs, feedback control mimicking the outer hair cells has been integrated into artificial cochlear sensory epithelia. Some research groups have used feedback control on piezoelectric sensors consisting of a single beam [42,43,44,45]. Our research group has developed a prototype of an artificial cochlear sensory epithelium capable of frequency selectivity, conversion of mechanical and electrical signals, and feedback control by mimicking the functions of the basilar membrane, the inner hair cells, and the outer hair cells in the cochlea [29]. This artificial cochlear sensory epithelium is intended for use by patients with sensorineural hearing loss. The device, which is composed of a trapezoidal membrane, contains both control and recognition electrodes. The recognition electrodes are used to measure the local electrical output caused by the oscillation of the device due to the piezoelectric effect. Local electrical stimuli are then applied via the control electrode to generate local oscillations for feedback control. When a sound stimulus is applied to the device, the resonance position oscillates with a large amplitude. Application of this local electrical stimulus at the resonance position superimposes the control oscillation with that of the sound stimulus. In this way, the oscillation amplitude can be amplified or dampened meaning the oscillation of the device can be controlled. However, it took 52.0 s to control the oscillation at the resonance position [29]. This is because the resonance position is determined based on the oscillation amplitudes for all of the electrodes, which are measured by a laser Doppler vibrometer while an automatic stage is moved to scan the measurement position. The feedback process must be shortened for practical use, because the device must follow frequency changes in daily life.

In the present study, improvements were made on results from our previous studies by implementing more rapid signal processing methods. That is, a novel high-speed feedback control mimicking the outer hair cells by implementing more rapid signal processing methods was experimentally demonstrated to integrate the amplification of the oscillation in the basilar membrane into an artificial cochlear sensory epithelium. The oscillation stimulus in this study was applied to the boundary of the device, whereas in the previous study on this device [29], the sound stimulus was produced by a speaker. It is relatively difficult to control the directivity of the speaker when a large area should be stimulated by a sound propagating through the air. On the other hand, applying an oscillation stimulus at the boundary is equivalent to applying a uniform external pressure and enables the evaluation of the oscillation characteristics under the same conditions for all measurement points. Moreover, the device from our previous study was improved [29] by fabricating a piezoelectric bimorph membrane to produce the electrical signals large enough to measure the resonance position. The frequency characteristics of both the oscillation amplitude and the electrical output were systematically, simultaneously, and quantitatively obtained. In this study, the resonance position was determined from the electrical output instead of using the previous method [29] based on the oscillation amplitude; this means that the time required to move the measurement position with the stage can be eliminated. In fact, the feedback control in the present study took 800 ms, equivalent to a reduction of 98.5% relative to that in the previous study. At the identified resonance position, where small discrete electrodes for both input and output are controlled by the local electrical stimulus, the local oscillation amplitude is magnified mimicking the functions of the outer hair cells. The neighboring two electrodes are also electrically controlled to reduce the amplitude, resulting in a high Q-value. Such techniques of high-speed feedback control can be applied to the development of fully implantable cochlear implants with a broad frequency selectivity and a wide dynamic range for future clinical applications.

## 2. Experimental Methods

### 2.1. Design and Fabrication of the Artificial Cochlear Sensory Epithelium

Research over the last several decades has focused on not only the kinds of fully implantable cochlear implants described in Section 1 but also models [46,47,48,49] that describe cochlear mechanics. To develop fully implantable cochlear implants, results obtained using these models should be considered when developing devices. However, there are limitations to representing actual cochlear behavior with this type of mechanical system. As a first step in the present study, the functions of the basilar membrane and the inner and the outer hair cells are considered. In particular, a feedback procedure mimicking the function of the outer hair cells was also developed in this study.

Following our previous study [29], Figure 1a,b show a schematic and a photograph, respectively, of the artificial cochlear sensory epithelium developed in this study to perform the functions of the outer hair cells using electrical feedback. In comparison with our previous device, the present device has a geometric configuration that is optimized for setting the bimorph PVDF membrane, as shown in Figure 1c making it more efficient as a piezoelectric actuator or sensor. The PVDF trapezoidal membrane has a length of 25 mm in the *x*-direction, and widths in the *y*-direction of 4 and 8 mm at *x* = 0 and 25 mm, respectively. The present device is a model in which the cochlea is unrolled and straightened as a first step for the development of fully implantable cochlear implants. This straightened model is very familiar well-used and is authorized in the research field of the inner ear and the hearing system. From the viewpoint of bioengineering, the present device is for an in vitro system. A future device should be miniaturized to almost the same dimensions with the basilar membrane in order to implant inside the cochlea, as summarized in Appendix A. In general, when a single tone is applied to the device, the position with the maximum oscillation amplitude is determined based on the variation in the local resonance frequencies, as shown in Figure 1d. The resonance frequency is dependent on the width in *y*-direction, and the resonance position varies with the input frequency; this phenomenon is called frequency selectivity. When the resonance position is vibrated, the fundamental oscillation mode is mostly observed in the *y*-direction, as shown in Figure 1e. In both the developed artificial cochlear sensory epithelium and the biological basilar membrane, the wavelength of the membrane oscillations produced by external sound stimuli in the *x*-direction is longer than that in the *y*-direction. It can be assumed that a wave propagating in the *x*-direction is negligibly small, and estimation of the resonance characteristics of a trapezoidal membrane has been achieved using a well-known simple method involving a beam vibrating in the fundamental mode under fixed conditions at both ends of the beam in the *y*-direction [22]. In this physical and mathematical modeling approach, the width of the beam element in the *x*-direction does not need to be considered in the theoretical predictions.

As described in Section 1, piezoelectric materials are used in fully implantable cochlear implants. That is, the electrical output produced by the piezoelectric effect is measured to determine the local resonance position, and local electrical stimuli are applied to superimpose the oscillation produced by the inverse piezoelectric effect on that produced by the external force. Therefore, recognition and control electrodes composed of Al are deposited on the PVDF membrane, as shown in Figure 1. The recognition electrodes are triangular in shape and are attached near the fixed boundary because the electrical signals generated by strain at the fixed boundary are measured to avoid getting those with opposite sign. Additionally, the control electrodes are inverted triangles meaning that they widen toward *x* = 0 mm to make the control electrodes distinguishable from the recognition electrodes. The fabrication process and conditions of the patterned electrodes are summarized in Table 1 and Figure 2. The Al electrodes are deposited with a thickness of approximately 50 nm on both sides of the PVDF membrane, which has a thickness of 80 μm (KF Piezo film, Kureha Corp., Tokyo, Japan). Note that the Al electrodes are negligibly thin relative to the PVDF membrane to reduce their effect on the membrane oscillation, so the local resonance frequency is determined by the physical properties and dimensions of the PVDF. A positive photoresist (AZ5214-E, Merck KGaA., Darmstadt, Germany) is spincoated on the upper Al electrode, as shown in Figure 2b. After a prebake, the photoresist is exposed to ultraviolet light using a maskless lithography tool (DDB-701-DL, Neoark Corp., Japan), as shown in Figure 2c. The exposed photoresist is then developed by immersing the membrane in developer solution (AZ300MIF, Merck KGaA., Darmstadt, Germany). The Al electrode, which is not covered by the photoresist, is simultaneously etched during the photoresist development, as shown in Figure 2d. The membrane is immersed in ethanol to remove the photoresist producing the patterned electrodes, as shown in Figure 2e. The lower Al electrode is also etched with the same process to obtain an electrode with a trapezoidal shape. The resulting membrane with patterned electrodes is bonded to another membrane with a thickness of 40 μm using an electric conductive adhesive (Dotite, D-550, Fujikura Kasei Co., Ltd., Tokyo, Japan) and an epoxy adhesive (CA-191, Cemedine Co., Ltd., Tokyo, Japan), as shown in Figure 1c. Note that a thickness of the adhesive layer is the order of 1 μm achieving by applying high pressure and the effect on the oscillation characteristics of the trapezoidal membrane must be negligibly small. In general, a bimorph which consists of membranes with the same thickness generates a larger electrical output, but the frequency range of the device fabricated with the two membranes with a thickness of 80 μm did not coincide with that of the previous study [29]. Therefore, dimensions of the present device are adjusted so that the frequency range overlaps and the time response of the feedback control is compared. The lower 40 μm membrane is used not only to shift the neutral axis but also to bring the lower electrode of the top membrane into contact with the upper electrode of bottom membrane. In this way, the potential difference between these upper and lower electrodes of the top membrane can be measured. The membrane was attached to a jig with a trapezoidal slit fabricated with a 3D printer (Objet Eden 260S, Stratasys Ltd., MN, USA), using double-sided tape (No.500, Nitto Denko Corp., Osaka, Japan). The electrical output of the bimorph membrane was simulated using COMSOL Multiphysics 5.3 (COMSOL Inc., Stockholm, Sweden), and was found to be larger than that of a single membrane; however the simulation results are omitted here.

### 2.2. Experimental Setup for Measurements and Feedback Control

To control the experimental apparatus and analyze the experimental results, a data acquisition (DAQ) system was used, as shown in Figure 3a. This DAQ consisted of a controller (NI PXIe-1082, NI PXIe-8840, National Instruments, Austin, TX, USA), a multichannel analog output module (NI PXIe-6738, National Instruments, Austin, TX, USA) for electrical stimulation, and a vibration module (NI PXIe-4492, National Instruments, Austin, TX, USA) for data analysis. The DAQ was also used to construct the feedback control system, and the detail control process is described in Section 2.3. A laser Doppler vibrometer (LDV; AT3600/AT0023, Graphtec Corp., Tokyo, Japan) was used to measure the local vibration of the patterned electrodes, and the steady-state oscillation z(t)=Zsin(2πft) was given particular focus. It was established that the DAQ system and the LDV could respectively measure electrical and vibration signals of $O(10^{-6})$ V and $O(10^{-10})$ μm with a time resolution of 200 kHz. The device was set on an xy automatic stage (SGSP26-150/SGSP20-35, Sigma Koki Co., Ltd., Japan) to control the oscillation measurement position. In this study, the following experiments were performed.

Each electrode deposited on the artificial cochlear sensory epithelium was oscillated via the inverse piezoelectric effect by applying a local electrical stimulus $V_{\mathrm{in}}\left(t\right)=V_{\mathrm{local},\mathrm{elec}}\sin\left(2\pi f_{\mathrm{local},\mathrm{elec}}t\right)$ to the control electrodes. The steady-state oscillation of $z\left(t\right)=Z\sin\left(2\pi f_{\mathrm{local},\mathrm{elec}}t\right)$ for each electrode was measured by the LDV, and the frequency characteristics of the oscillation amplitude *Z* were analyzed to observe the frequency selectivity of the device, as described in Section 3.1.The displacement $z\left(t\right)=Z\sin\left(2\pi f_{\mathrm{mech}}t\right)$ and the electrical output $V\left(t\right)=V_{\mathrm{out}}\sin\left(2\pi f_{\mathrm{mech}}t\right)$ generated by the piezoelectric effect were simultaneously measured with external mechanical oscillations applied to the fixed trapezoidal boundary with a constant acceleration amplitude $a_{B.C.}\left(t\right)=a_{B.C.}\sin\left(2\pi f_{\mathrm{mech}}t\right)$. The frequency characteristics of the amplitudes *Z* and $V_{\mathrm{out}}$ were evaluated, as described in Section 3.2.Based on the output signals from the device, the oscillations of the artificial cochlear sensory epithelium $z\left(t\right)=Z\sin\left(2\pi f_{\mathrm{mech}}t\right)$ were controlled, as described in Section 3.4. Testing this feedback control approach was the main purpose of this study.

In our previous study [29], a speaker was used to actuate the device when feedback control was performed. However, it is difficult to control the directivity of the speaker when a large area is stimulated by the sound through an air medium as described above. Therefore, the part connecting the speaker with the device was used to locally actuate each electrode. In this study, the oscillation system shown in Figure 3b was developed to oscillate the device, and the fixed trapezoidal boundary was oscillated with a constant acceleration amplitude $a_{B.C.}\left(t\right)=a_{B.C.}\sin\left(2\pi f_{\mathrm{mech}}t\right)$, which means that the device was stimulated by uniform pressure in the xy plane. The oscillation system consists of the stage, an O-ring, a plate, and an actuator. The actuator oscillates the plate, and then the device is oscillated. The electrical output is produced due to the mechanical oscillation of the device, but the electromagnetic waves produced by the actuator cause an unexpected fluctuation. Therefore, the electrical shields surround the actuator and the probes to reduce the noises. The O-ring is used to make the fixed boundary show spatially uniform oscillations. Note that the acceleration of the plate is essentially a function of *x*, *y*, *t*, and $f_{\mathrm{in}}$; however it was confirmed that the acceleration of points along the fixed boundary were nearly spatially constant, and the local variation in the xy plane is neglected. The absolute value of the maximum acceleration varies as a function of the input frequency, as shown in Figure 4 as a result of the oscillation characteristics of the base plate. Therefore, the voltage input to the actuator is optimized to keep $a_{B.C.}$ constant with varying $f_{\mathrm{mech}}$. In all experiments, a sinusoidal external force was applied to the device, and the waveform data were measured and analyzed. To precisely extract the frequency characteristics, a fast Fourier transform (FFT) was performed on the experimental data, and the frequency components of the input signal were extracted.

### 2.3. Noteworthy Improvement of Feedback Control

In this study, the improved feedback control system shown in Figure 3a was used to realize a much higher temporal performance in the modulation responses. First, the method used in our previous study [29] is explained here. In this previous study, a prototype artificial cochlear sensory epithelium with six pairs of electrodes that performs the functions of recognition and control in a manner similar to the device developed in the present study was fabricated for the first time. When external sound stimuli are applied to the device, the local displacement at each electrode is measured by the LDV with a moving stage in the *x*-direction to identify the local resonance position, and the oscillation amplitudes near the electrodes are quantitatively compared. The target electrode, at which oscillations occur with the maximum amplitude, is determined automatically. A local electrical stimulus is then applied to the target electrode to superimpose the resonant oscillations with the oscillations generated by the external force. To amplify the oscillations, vibrations with the same phase are also applied. Moreover, the neighboring electrodes are also stimulated by vibrations of the opposite phase to damp the oscillations and improve the dynamic range. However, the feedback control procedure described above takes approximately 50 s because moving the measurement position to scan the six electrodes in the *x*-direction adds unnecessary time. This is the disadvantage of our previous feedback control scheme that we sought to improve here. Herein, to improve the determination of the electrode with the maximum oscillation amplitude, the piezoelectric output signal from the artificial cochlear sensory epithelium was used to identify the resonance position. In the previous study, the frequency characteristics of the electrical output from the device were not clearly observed, and only the relationship between the electrical output and the oscillation amplitude was clarified. In the present study, the device was developed to have a bimorph membrane to enable the observation of the frequency characteristics of the electrical output, as described in Section 3.2. Furthermore, the resonance position was determined by simultaneously measuring the electrical output from each electrode. These are the points of improvement achieved in the present novel scheme of feedback control, and they were realized by using a bimorph membrane and local electrodes. With this setup, it is no longer necessary to move the measurement position to scan multiple electrodes, and the revised design to drastically reduces the time required for feedback control. In the present system, an amplifier or a buffer are not used, and the gain is not introduced to the feedback control.

## 3. Experimental Results and Discussion

### 3.1. Frequency Selectivity of the Artificial Cochlear Sensory Epithelium Measured with the Inverse Piezoelecric Effect

The oscillation of the trapezoidal membrane was induced via the inverse piezoelectric effect by applying an input voltage of $V_{\mathrm{in}}\left(t\right)=V_{\mathrm{local},\mathrm{elec}}\sin\left(2\pi f_{\mathrm{local},\mathrm{elec}}t\right)$ to the control electrode shown in Figure 1a. The basic oscillation characteristics of the membrane were successfully estimated with the developed model. This method achieved better performance than the previous method using sound stimuli from the speaker system in terms of the accuracy, stability, and reproducibility of the data because of the shortcut of the surrounding medium such as an air or lymphatic fluid [35]. The experimental run was performed by automatically varying $f_{\mathrm{local},\mathrm{elec}}$ from 3.0 to 12.0 kHz at intervals of 0.1 kHz using LabVIEW software (2017, National Instruments, Austin, TX, USA), and the oscillations were measured with a sampling rate of 100 kHz and an average duration of 0.1 s for each plot.

Figure 5 shows the local oscillation amplitude *Z* for the *n*th electrode plotted against the frequency $f_{\mathrm{local},\mathrm{elec}}$ of the applied voltage to demonstrate the inverse piezoelectric effect for an input voltage amplitude of $V_{\mathrm{local},\mathrm{elec}}$ = 10 V. The plotted data were analyzed by taking the FFT of the raw time-series data obtained by the LDV as described in Section 2.2. This figure shows the resonance characteristics of the trapezoidal membrane at the electrodes shown in Figure 1a. Large peaks can be observed in the plots for the 3rd, 5th, and 10th electrodes (n=3,5,10), and the frequencies at these peaks represent the local resonance frequency $f_{r,\exp}^{\left(n\right)}$ for the *n*th electrode. Here, the linear theory of beam vibration is introduced to validate the frequency selectivity of the trapezoidal membrane. The trapezoidal membrane is modeled as a set of fixed-end beams with lengths l(x) in the *y*-direction that change continuously along the *x*-direction. The theoretical resonance frequency $f_{r,\mathrm{theory}}^{\left(n\right)}$ for the fundamental oscillation mode is expressed by the following equation based on the Euler–Bernoulli theory for transverse vibrations of a beam:(1)$$f_{r,\mathrm{theory}}^{\left(n\right)}=\frac{22.37}{2\pi l^{2}\left(x\right)}\sqrt{\frac{Eh^{2}}{12\rho }},$$
where *E* is the modulus of elasticity, ρ the mass density, and *h* the thickness of the beam. In the proposed device, *E*, ρ, and *h* are constant along the *x*- and *y*-directions, and $f_{r,\mathrm{theory}}^{\left(n\right)}$ is mainly determined by l(x). Therefore, the resonance frequency decreases with increasing electrode number *n*, and the resonance frequencies of electrodes 1, 4, and 7 can be determined based on this principle despite the presence of multiple peaks, as shown in Figure 5 and Table 2. From Equation (Equation 1), the following relationship can be obtained:(2)$$f_{r,\mathrm{theory}}^{\left(n\right)}l^{2}\left(x\right)=C,$$
where *C* is a constant. Based on Equation (Equation 2), the theoretical the experimental results are validated. However, it is difficult to determine the elastic modulus *E* of the PVDF bimorph membrane which varies depending on the material processing. Furthermore, the value also varies among the literature, e.g., *E* is 3.0–3.6 GPa in Ref. [50]. By substituting the values of *E* = 3.6 GPa, ρ = 1780 $\mathrm{kg}/m^{3}$, and *h* = 120 μm into Equations (Equation 1) and (Equation 2), we obtained *C* = 175 ${\mathrm{mm}}^{2}\;\mathrm{kHz}$. On the other hand, by substituting the values of $f_{r,\exp}^{\left(n\right)}$ obtained by the experimental results into Equations (Equation 1) and (Equation 2) for electrodes *n* = 1, 3, 4, 5, 6, and 10, an averaged value of *C* = 175 ± 10.6 ${\mathrm{mm}}^{2}\;\mathrm{kHz}$, which agreed well with the theoretical value, was obtained, and the deviation was approximately 6.06%. In this way, the fixed-end beam model for the prediction of the local resonance frequency was validated for practical applications, and the frequency selectivity of the trapezoidal membrane was quantitatively demonstrated.

The data from electrodes 8, 9, 11, and 12 are omitted here because they could not be reasonably estimated from the standpoint of accuracy and reproducibility in the experimental runs. The control electrodes often became disconnected even though they are connected after photolithography due to the dynamic behavior of the device in response to the repeated high-frequency oscillations. Insufficient hardening of the photoresist or the too long development made the electrodes with unexpected precision. The stress concentrates on the smaller part, which may cause the disconnection of the electrodes. In consideration of the data from electrode 10 shown in Figure 5, the amplitude of the membrane becomes relatively large with increasing *x* as a result of the increasing length l(x) of the virtual beam elements in the *y*-direction. The data from electrodes 2 and 6 are also omitted because of an unexpected error in the resonance frequency analysis and challenges with reproducibility resulting from difficulties with the fabrication process, specifically adhesion bonding between the bimorph PVDF membrane and the acrylic plate. The repeated high-frequency oscillations cause damage to the fixed end, and the results are not reproducible since the the experiments were performed for long periods. As described above, the resonance frequency for each virtual beam was successfully predicted by the well-known Euler–Bernoulli theory. As a result of the quality of the data from electrodes 3, 5, and 10, these three cases are treated as the expected and ideal behavior of the membrane, as described by the linear theory for the fundamental oscillation mode. It was found that the resonance frequency decreases with increasing electrode number, which corresponds to the width of the trapezoidal membrane in the *y*-direction. This trend constitutes theoretical and experimental proof that the trapezoidal membrane is capable of frequency selectivity, where the trapezoidal membrane acts as an analog FFT. For electrodes 1, 4, and 7, the resonant oscillation frequency does not produce the maximum oscillation amplitude although the resonance frequencies $f_{r,\exp}^{\left(1\right)}$, $f_{r,\exp}^{\left(4\right)}$, and $f_{r,\exp}^{\left(7\right)}$ do yield lower peaks. In these cases, the resonance frequency can certainly be predicted reasonably, but the total amplitude is determined by other phenomena. For example, electrode 4 certainly resonates at $f_{\mathrm{in}}=f_{r,\exp}^{\left(4\right)}$ despite $f_{r,\exp}^{\left(5\right)}$ yielding the maximum amplitude at this electrode. This is likely due to the strong effect of oscillations at the neighboring electrode 5, which is the result of the continuous dynamics of the trapezoidal membrane, i.e., the virtual beams are connected to each other in the *x*-direction in the trapezoidal membrane and thus do not show independent motion. In our previous study, it was found that edge effects at *x* = 0 and 25 mm also have a strong effect on the local oscillations near electrodes 1 and 12. From Figure 5, it was concluded that the resonance frequency near the electrode can be well predicted by the above mentioned linear theory. However, the prediction of the local amplitude is difficult because of the complex phenomena governing the behavior of continuous media.

As shown in Figure 5, we can obtain a lot of peaks corresponding to higher oscillation mode and the effect of interactions between the electrodes, where the detailed mechanism should be fully investigated in future work. Furthermore, the outstanding peaks show the resonance frequency of the fundamental oscillation mode. The experimental results are validated based on the Euler–Bernoulli theory for transverse vibrations of a beam. The accumulation of the basic data of resonance provides the basic knowledge and useful clues for future research.

### 3.2. Measurement of Output Electrical Signal by Applying the External Mechanical Oscillations

The local oscillation amplitude for the patterned electrodes and the frequency response of the output electrical signal obtained from the electrodes were then measured under an applied external mechanical vibration stimulus. This experiment was conducted mainly for the accumulation of practical and basic data on the membrane when it is applied as an electrical power generator. To quantitatively evaluate the oscillation amplitude of the trapezoidal membrane, the voltage input to the actuator was optimized so that the acceleration of the vibration of the fixed boundary of the membrane was nearly constant at $a_{B.C.}$ = 0.2 $m/s^{2}$. This value corresponds to a sound pressure level of 67.6 dB, which is almost the same as that for an ordinary conversation. Note that the acoustic pressure produced by the actuator is negligible small in comparison with the mechanical vibration. It was confirmed that the electrical output was the order of 1 µV which is much smaller than the experimental results shown later when only the acoustic pressure with the resonance frequencies is applied to the device. Sound stimulation is indirect and is through the medium of the air, whereas the vibration one through the solid base plate is physically direct contacted, so the energy transfer seems to be a different order of magnitude.

Figure 6 shows the measurement results for the local amplitude *Z* of the membrane oscillation induced by a mechanical stimulus applied to the base plate and the signal $V_{\mathrm{out}}$ of piezoelectric output for various mechanical stimulus frequencies $f_{\mathrm{mech}}$ at the locations of electrodes 3, 4, 5, and 7. Each experimental plot in the Figure 6 shows the amplitude averaged over five experimental runs, and the deviations are relatively small (within 2.63%), as described by the error bars in the figure. In the case of electrode 3, there is the clear peak in $V_{\mathrm{out}}$ at the resonance frequency $f_{r,\exp}^{\left(3\right)}$, which was determined via the inverse piezoelectric effect as shown in Figure 5, even though multiple peaks were observed. In general, the electric output $V_{\mathrm{out}}$ is proportional to the strain as a result of the piezoelectric effect and should be maximized when the membrane is locally resonating. This relationship has been reported in our previous study [35]. In the present device, the electrodes were deposited near the fixed trapezoidal boundary, and the electrical output generated by the strain on the boundary was successfully measured at electrode 3. However, for the other electrodes, the value of $f_{\mathrm{mech}}$ that produces the maximum *Z* does not coincide with these resonance frequencies $f_{r,\exp}^{\left(n\right)}$. In the local frequency range around the resonance frequencies $f_{r,\exp}^{\left(4\right)}$, $f_{r,\exp}^{\left(5\right)}$, and $f_{r,\exp}^{\left(7\right)}$ shown in the inset of each set of results, *Z* and $V_{\mathrm{out}}$ are maximized at the resonance frequency for the corresponding to electrode, as shown in Table 2. Therefore, the oscillation amplitude *Z* and the output voltage $V_{\mathrm{out}}$ at a given electrode are strongly correlated when the trapezoidal membrane oscillates at the corresponding resonance frequency.

As shown in Figure 6, with the exception of $f_{r,\exp}^{\left(3\right)}$, the input frequency $f_{\mathrm{mech}}$ that produces the maximum output voltage and amplitude does not correspond to the resonance frequency $f_{r,\exp}^{\left(n\right)}$. For all experimental conditions when $f_{\mathrm{mech}}$ is close to $f_{r,\exp}^{\left(3\right)}$, the voltage and amplitude were maximized. As shown in Figure 5, because the oscillation amplitude for electrode 3 at $f_{r,\exp}^{\left(3\right)}$ is very large compared to that at other electrodes, the oscillation of electrode 3 affects the other electrodes. This is readily observable in Figure 6; peaks in $V_{\mathrm{out}}$ are present near $f_{\mathrm{mech}}=f_{r,\exp}^{\left(3\right)}$ even for other electrodes. As discussed above, it is difficult to reproducibly and uniformly fix the boundaries of the membrane. Thus, it is necessary to construct a novel scheme to recognize the resonance location recognition; this was the main purpose of this study and is presented in the next section. Figure 6 shows that the peak of oscillation amplitude at the resonance frequency agrees with that of electrical output and the resonance frequency can be determined by the measurement results of the electrical output. It is clarified that the time for the determination of the resonance position can be reduced because the movement of the stage for the measurement of the oscillation amplitude [29] is not required and the high-speed feedback control is achieved.

Figure 7 shows the output electrical signals $V_{\mathrm{out}}$ for various mechanical input frequencies $f_{\mathrm{mech}}$ for electrodes 3, 4, 5, and 7. When a mechanical external oscillation with $f_{\mathrm{mech}}=f_{r,\exp}^{\left(n\right)}$ was applied, the output at electrode *n* shows a larger peak than that at other electrodes. For example, when the device is mechanically oscillated with $f_{\mathrm{mech}}=f_{r,\exp}^{\left(3\right)}$, $V_{\mathrm{out}}$ for electrodes 3, 4, 5, and 7 are 0.280, 0.0402, 0.0435, and 0.0813 mV, respectively. Therefore, $V_{\mathrm{out}}$ for electrode 3 is the highest among the electrodes, and this electrode can be determined as corresponding to the local resonance position. These results strongly suggest that the recognition of local resonance in the feedback control process based on the output voltage of each electrode is more suitable than that our previous scheme.

### 3.3. Challenges in the Fabrication Process and Oscillation Measurements for the Continuous Trapezoidal Membrane

In Section 3.1, the data from electrodes 8, 9, 11, and 12 are omitted mainly due to the disconnection. For future clinical use of the artificial cochlear sensory epithelium, the fabrication process must be validated and the optimized. First, materials of the electrodes should be selected from the viewpoint of the durability and the corrosion resistance. In the present study, aluminum electrodes are deposited on the piezoelectric membrane because they are rather resistant to corrosion and stick well to PVDF membrane. In addition, the electrodes are used to stimulate the auditory nerve for sensorineural hearing loss. The motion of the ions in the lymph fluids is actuated by the electrical stimuli via the electrodes. Therefore, the metal such as gold is useful to attract ions and to stimulate the auditory nerve cells, where the further investigation of the electrochemical phenomena remains as future work. Second, dimensions of the electrodes should be determined based on a stress analysis using the well-known commercial code. The disconnection of the electrodes occurred mainly at the leader lines, where it is very important and difficult problems in MEMS experiment. Therefore, the stress at a trapezoidal fixed boundary should be calculated and the configuration of the patterned electrodes will be improved. Finally, the photolitography process should be reconsidered to fabricate the feasible electrodes. The difference of coefficient of thermal expansion causes residual stress in the prebake. The chemical etching is used to fabricate the patterned electrodes and they are often made in different dimensions, e.g., the lead lines are a bit smaller than the design, which causes disconnections. A thickness of the photoresist and the time of the development and etching should be optimized to improve the reproducibility of the device.

Although 12 pairs of electrodes are deposited on the PVDF membrane, only 4 electrodes could stably measure the local electrical signals generated by the mechanical external force with high accuracy and high reproducibility. In fact, beam type sensors can be more easily fabricated and are widely used as described in Section 1. However, a beam array responds discretely to sound frequencies, whereas a trapezoidal membrane responds continuously. Therefore, the present study using a continuous trapezoidal membrane is significant from the viewpoint of the natural hearing. In the study of fully implantable cochlear devices, microelectrodes have commonly been deposited on piezoelectric membranes [22,23,25]. Shintaku et al. [22] fabricated a devices with 24 electrodes and Jung et al. [23,25] used 12 or 13 electrodes to measure the electrical output. Unfortunately, not all electrodes in our previous device [22] showed the clear relation between the input frequency and the electrical signal. In general, it is not simple to successfully deposit electrodes on membranes made of various materials or to measure electrical signals on the order of microvolts. Additionally, when the fixation at the boundary is insufficient because of the accuracy of the trapezoidal slit or the bonding of the membrane, the boundary conditions may be more akin to a simple support than a fixed end, as shown in Figure 8. The boundary conditions for the fixed end shown in Figure 8a and that for the simple support shown in Figure 8b can be described by
(3)$$z=\frac{\partial z}{\partial y}=0\;\mathrm{and}\;z=\frac{\partial^{2}z}{\partial y^{2}}=0,$$
respectively. In general, the electrical output generated by the deformation of the device is proportional to the stress, which is linearly related to the bending moment, so the electrical output is expected to decrease when the boundary conditions close to those for a simply supported beam. Considering the results given in Table 2 in Section 3.1, the constant values expressed by Equation (Equation 2) decrease with increasing the electrode number from electrodes 4 to 10. The resonance frequency decreases when the length increases or the beam is simply supported. Although a quantitative evaluation of the fixation was not performed because of the complex adhesion phenomena, it is difficult to fix the membrane with a uniform force. From the above discussion, the fabrication process for the electrodes and the fixation of the device must be optimized for future clinical applications.

### 3.4. Feedback Control of the Oscillation in a Trapezoidal Membrane

The feedback control of the oscillation amplitude using the control electrodes for the inverse piezoelectric effect shown in Figure 1a was performed based on the results obtained in Section 3.2. Feedback electrical stimulation that amplifies the oscillations of the electrode with the maximum oscillation amplitude, which is identified from the output voltage, was locally applied based on the recognized resonance frequency. Electrical stimulation was also applied to reduce the oscillation amplitude for the neighboring electrodes on both sides of the target electrode. The fixed boundary was oscillated with an acceleration of $a_{B.C.}$ = 0.2 $m/s^{2}$ and a feedback electrical stimulus with a magnitude of $V_{\mathrm{local},\mathrm{elec}}$ = 10 V was locally applied to the control electrodes. To confirm the effect of the feedback control mechanism constructed in this study, the controlled oscillation amplitude of the trapezoidal membrane *Z* was measured, and the feedback control results are shown in Figure 9. This figure shows the oscillation amplitude *Z* of the membrane plotted against the electrode number *n* with and without feedback control applied to the system. In the case without the feedback control, only the mechanical external vibration is applied to the base plate. When the mechanical oscillation without feedback control at the resonance frequencies is applied to the fixed boundary, electrodes 3 and 7 have the maximum amplitude even though the case of electrodes 4 and 5 show other smaller peaks at non-resonant electrodes. These results are representative of the other oscillation modes of the trapezoidal membrane that have been reported in previous studies [22,23,24,25,27,28,29] and provide evidence that trapezoidal membranes can perform the function of frequency selectivity. The resonance position is determined by measuring the local electrical output $V_{\mathrm{out}}$ in the feedback process, and the results obtained with the proposed method of feedback control are also shown in Figure 9. The oscillation amplitudes at the resonance positions were successfully magnified by applying local electrical stimuli via the control electrodes. However, the oscillation amplitudes for the uncontrolled electrodes 6, 8, and 10 also increased, and Figure 9b shows that the oscillation amplitude for electrode 3 exceeds that of electrode 4 when $f_{\mathrm{mech}}=f_{r,\exp}^{\left(4\right)}$. This result is likely mainly caused by the oscillation mode in the *x*-direction. The phenomena governing the artificial membrane behavior may be difficult and complex in comparison with biological ones because the fixed boundary conditions in the *x*-direction and the wave reflection result in the formation of a standing wave in the artificial trapezoidal membrane, whereas traveling waves in the organ of Corti in the human cochlea propagate along the direction of rotation of the basilar membrane. Furthermore, this behavior also depends on the method of the vibrating membrane to provide an external input mimicking a sound stimulus. The main purpose of this study was to control the local oscillation amplitude of the membrane in a manner mimicking the function of the outer hair cells. Thus, the key advancements achieved here are (1) a method of identifying the resonant location, which is expressed by a discrete electrode number, using the output voltage, and (2) the ability to perform instantaneous control more effectively than in our previous work based on the recognition by scanning the measurement of the oscillation amplitude before applying feedback control [29]. The feedback control was originally applied to our device to amplify the electrical output, but only oscillation amplitude data after the feedback control were obtained. This is mainly because producing electrical stimuli with the control electrodes may generate unexpected electromagnetic waves, meaning the electrical output after the feedback control cannot be measured with sufficient enough accuracy. Therefore, a countermeasure such as a Faraday cage must be fabricated to prevent this type of noise. In the previous study [29], the time required to achieve feedback control of the target electrodes was approximately 50 s, approximately 48 s of which was spent measuring the oscillation amplitude or moving the xy-stage to scan the electrodes as part of this measurement. The application of our novel principle of resonance recognition by measuring the output electrical signal without scanning the oscillation amplitude reduced the control time to approximately 0.8 s, drastically improving the speed of the feedback control. Of this 800 ms required for feedback control, it takes approximately 200 ms to measure the output electrical signal and apply electrical stimulation, whereas the remaining about 600 ms is the time for the DAQ processing. For more precise mimicking the outer hair cells from the viewpoint of the response time, the temporal resolution should be reduced to the order of milliseconds because the most common frequency range of the human conversation is about 1 kHz and a high-speed external central processing unit (CPU) or analog circuits should be introduced in the future to simplify device implantation by reducing the volume. However, the principle of the feedback control method proposed here is expected to be applicable for practical use. To quantitatively evaluate the present feedback control system, an evaluation index $Q_{z}$, defined as the ratio of the oscillation amplitude to the average amplitude of the two neighboring electrodes, is introduced as
(4)$$Q_{z}=\frac{Z^{\left(n\right)}}{(Z^{\left(n+1\right)}+Z^{\left(n-1\right)})/2}.$$

The values of $Q_{z}$ for electrodes 3, 4, 5, and 7 are given in Table 3. By estimating the effect of the feedback control on the oscillation amplitude, it was found that a steeper peak in the oscillation amplitude is closely connected with the Q-value, and the evaluation index $Q_{z}$ was found to increase by an average factor of 1.12 with the application of the feedback control method. These results strongly depend on the ratio of the amplitude with feedback control to that without. That is, the oscillation amplitude is at most 1.44 times amplified by the feedback control, and a greater amplification of the oscillation amplitude corresponds to more improvement of $Q_{z}$. In this study, electrical stimuli with a steady amplitude were applied to the control electrodes, and a constant-amplitude oscillation was superimposed on the oscillations driven by the external force. However, actual outer hair cells in the human cochlea control the oscillation of the basilar membrane in response to the loudness of the sound. Therefore, a feedback control principle based on the biomechanical model [51,52,53] should be integrated into the present device in the future to account for various magnitudes of the external force. We want to emphasize that all the data are automatically obtained by our DAQ system repeatedly to achieve the simultaneous measurements with the higher accuracy for the oscillation amplitude and the electrical output. Among the lot of following works based on our early concept of MEMS cochlea [54], it is noted that we have the advantage of the feedback control system mimicking the outer hair cells based on the simultaneous measuring techniques with high accuracy.

## 4. Conclusions

We propose a method of high-speed feedback control for an artificial cochlear sensory epithelium that mimics the function of the outer hair cells in the organ of Corti to realize high output signal amplification and an improved Q-value. The main results of the present paper can be summarized as follows.

An improved artificial cochlear sensory epithelium was developed on the basis of our previous device [29]. The device consists of a bimorph PVDF piezoelectric membrane to achieve frequency selectivity from the piezoelectric output and generate a higher voltage. A mechanical oscillation system that can apply a constant acceleration to the boundary of the trapezoidal membrane was used to measure the electrical signals from the device.The frequency characteristics of the oscillation amplitude and the electrical output were systematically and simultaneously measured, and it was found that they the showed peaks at the corresponding resonance frequency for each electrode. Thus, the measurement of the electrical output can be used to identify the resonance position.The adhesion of the PVDF membrane to the jig strongly affected the accuracy and reproducibility of the data, which was qualitatively discussed based on the linear theory of beam oscillation. When the boundary conditions were not fixed ends but simple supports, the resonance frequencies and the piezoelectric output generated by the strain were reduced. The validity of the model was qualitatively confirmed by comparing the theoretical values with the experimental results.Feedback control of the membrane oscillation was performed. The resonance position was determined by the amplitude of the electrical output, and the corresponding electrode was electrically stimulated via a nearby control electrode. The oscillation amplitude at the resonance position was successfully amplified in this way, and the sharpness of the vibration peak was improved in experimental tests. In comparison with the previously developed feedback principle [29], the speed of feedback control was drastically improved, with the control time reduced by approximately 99%.

## Figures and Tables

**Figure 1 micromachines-11-00644-f001:**
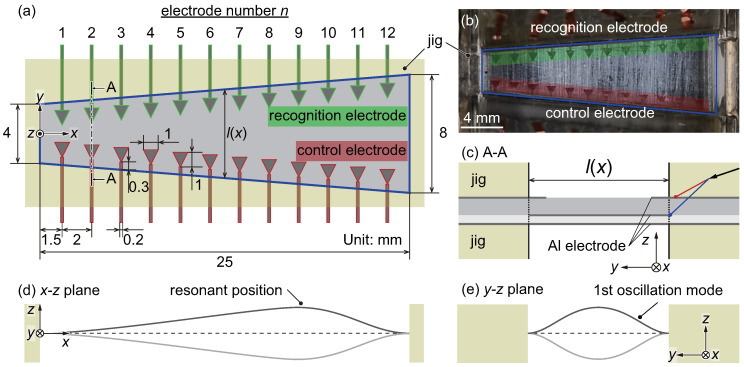
Artificial cochlear sensory epithelium fabricated in this study. (**a**) Schematic of the device. The piezoelectric membrane has a trapezoidal fixed boundary to mimic the shape of the basilar membrane. The device contains recognition electrodes to locally measure the electrical output, and control electrodes to locally oscillate the device via the inverse piezoelectric effect. (**b**) Photograph of the artificial cochlear sensory epithelium. (**c**) Side view of the membrane in the yz plane. The device is a bimorph membrane and consists of two polyvinylidene difluoride (PVDF) membranes with thicknesses of 80 and 40 μm. The 80 μm membrane contains deposited Al electrodes to measure electrical output and/or apply electrical stimuli. (**d**) Illustration of resonance in the xz plane. When a single tone is applied to the device, the resonance position is determined as the position with the maximum oscillation amplitude is observed. The resonance position varies depending on the frequency, providing a frequency selectivity function for the trapezoidal membrane. (**e**) Illustration of the oscillation mode in the yz plane. The local resonance frequency is dependent on the length in the *y*-direction, and the local vibration is described by the first oscillation mode.

**Figure 2 micromachines-11-00644-f002:**
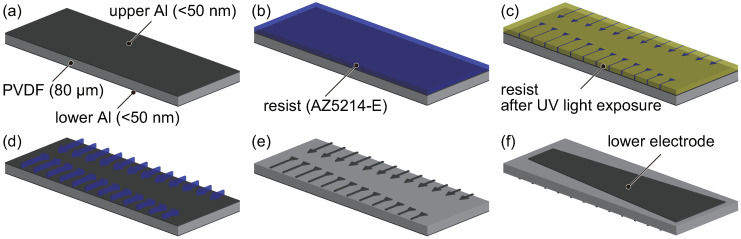
Fabrication process for the patterned electrodes deposited on the PVDF membrane. (**a**) Al electrodes are deposited on both the upper and lower sides of the PVDF membrane. (**b**) The upper electrode is spincoated with photoresist AZ5214-E. (**c**) The resist is exposed to ultraviolet light. (**d**) The exposed photoresist is developed using a developer solution. (**e**) The uncovered electrodes are etched during the development, and the photoresist is removed. (**f**) The lower electrode is prepared using the same process.

**Figure 3 micromachines-11-00644-f003:**
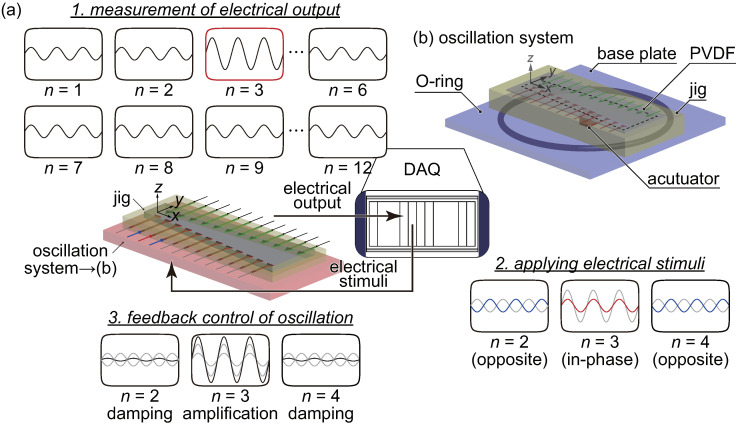
(**a**) Schematic of the feedback control system. A data acquisition (DAQ) system is used to measure the local electrical output produced by the oscillation of the device and to apply electrical stimuli. The device is actuated by the oscillation system shown in (b). (**b**) Schematic of the oscillation system. The oscillation system consists of a base plate, an O-ring, and an actuator. The PVDF membrane is fixed on a jig with a trapezoidal slit. The device is set on the base plate, which is oscillated by the actuator. The O-ring is used to uniformly oscillate the boundary of the device, and the amplitudes and phases of the boundary acceleration $a_{B.C.}\left(t\right)$ were made to be approximately constant at the multiple measurement points by optimizing the behavior of the actuator.

**Figure 4 micromachines-11-00644-f004:**
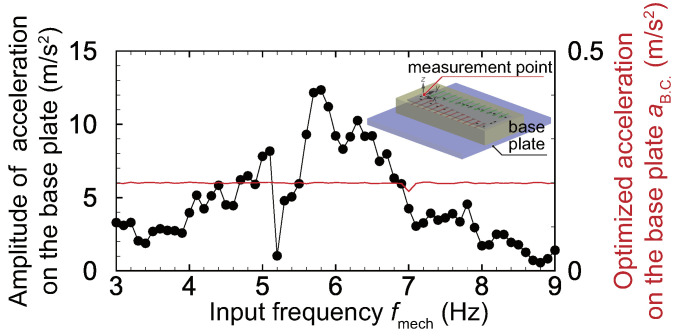
An absolute value of the maximum acceleration at x=y=0 mm plotted against the input frequency $f_{\mathrm{mech}}$ under an applied electrical stimulus of $V_{\mathrm{in}}\left(t\right)=V_{\mathrm{in}}\sin\left(2\pi f_{\mathrm{mech}}t\right)$, where $V_{\mathrm{in}}$ = 5 V. The maximum acceleration varied with the input frequency, so the electrical stimulus applied to the actuator was optimized to evaluate the frequency characteristic of the device by applying the appropriate oscillation. The optimized amplitude of the acceleration at the boundary is shown by the red solid line.

**Figure 5 micromachines-11-00644-f005:**
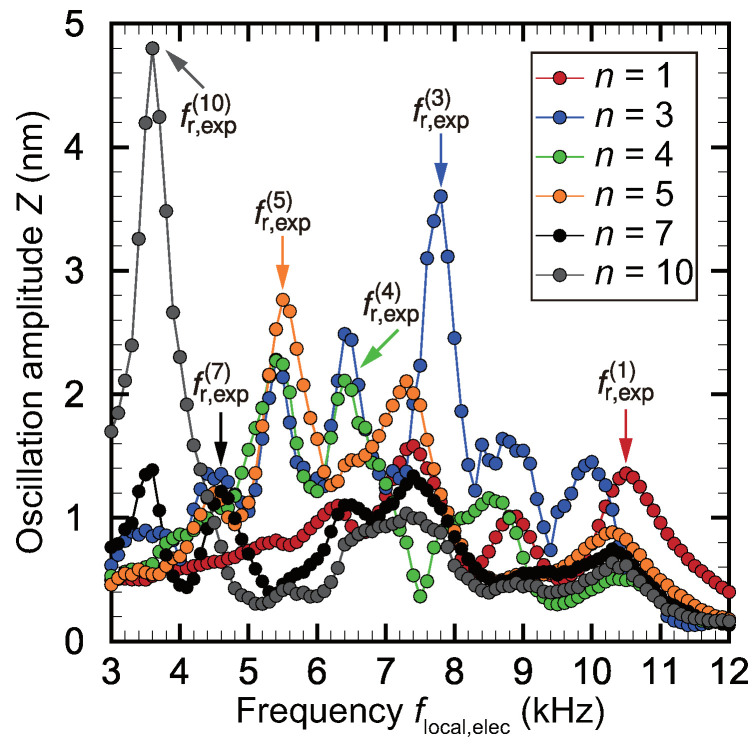
Local oscillation amplitude *Z* of each electrode measured via the inverse piezoelectric effect plotted against the input frequency of the local electrical stimulus $f_{\mathrm{local},\mathrm{elec}}$. Large peaks are present for electrodes 3, 5, and 10, and the location of these peaks can be identifies as the local resonance frequencies $f_{r,\exp}^{\left(n\right)}$ (n=3,5,10). However, it is difficult to determine the resonance frequencies for electrodes 1, 4, and 7, because they show no clear peaks as a result of their oscillation in response to neighboring resonance frequencies. Therefore, these resonance frequencies are determined by theoretical prediction. It is revealed that the present device has the frequency selectivity as the analog fast Fourier transform (FFT) device mimicking the biological basilar membrane in the cochlea due to the trapezoidal configuration, which is the key functions for fully implantable cochlear implants.

**Figure 6 micromachines-11-00644-f006:**
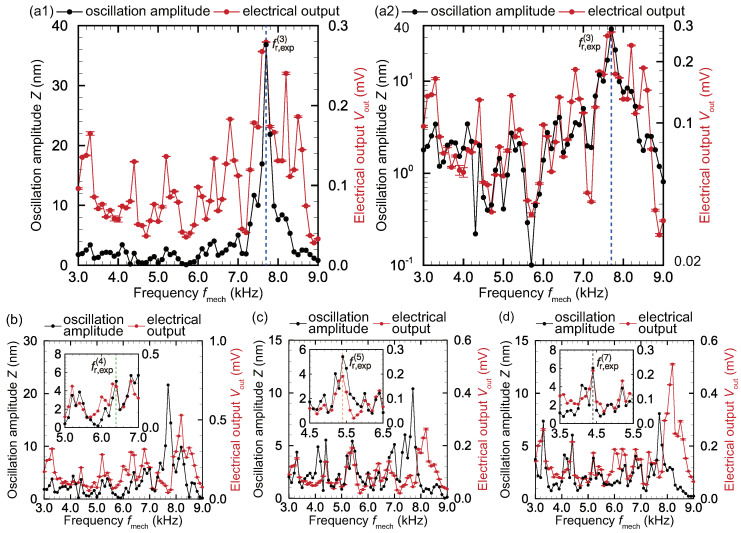
Frequency characteristics of the oscillation amplitude and the electrical output for electrodes (**a**) 3, (**b**) 4, (**c**) 5, and (**d**) 7, when the device was oscillated by the actuator. In the panel (a2), the oscillation amplitude and the electrical output are plotted in log scale. In the case of electrode 3, *Z* and $V_{\mathrm{out}}$ were maximized at $f_{r,\exp}^{\left(3\right)}$, and the resonance frequency could be determined by measuring $V_{\mathrm{out}}$. The insets in (b)–(d) show magnified views near the corresponding resonance frequencies $f_{r,\exp}^{\left(n\right)}$ (*n* = 4, 5, 7). At the resonance frequency, peaks are present in both the oscillation amplitude and the electrical output, and the electrical output is generated by the strain produced at resonance. The frequency selectivity is shown by both the oscillation amplitude and the electrical output and the time for the determination of the resonance position can be reduced by measuring the electrical output.

**Figure 7 micromachines-11-00644-f007:**
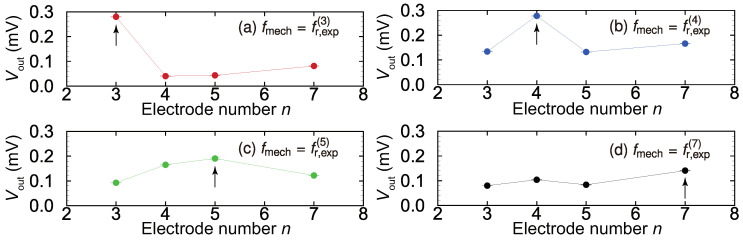
Electrical output $V_{\mathrm{out}}$ plotted against electrode number *n* for various resonance frequencies: (**a**) $f_{\mathrm{mech}}=f_{r,\exp}^{\left(3\right)}$, (**b**) $f_{\mathrm{mech}}=f_{r,\exp}^{\left(4\right)}$, (**c**) $f_{\mathrm{mech}}=f_{r,\exp}^{\left(5\right)}$, (**d**) $f_{\mathrm{mech}}=f_{r,\exp}^{\left(7\right)}$. $V_{\mathrm{out}}$ from the *n*th electrode is maximized at $f_{r,\exp}^{\left(n\right)}$ (*n*=3, 4, 5, 7) and is applied as indicated by the black arrows. Therefore, the resonance position can be determined from the electrical output, which is the advantages of the proposed method.

**Figure 8 micromachines-11-00644-f008:**
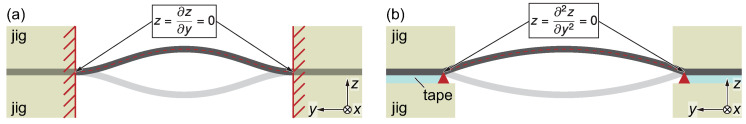
Schematic illustration of the boundary. (**a**) Fixed condition based on beam theory. The displacement and gradient of the membrane at the boundary are equal to zero. (**b**) Possible state of the actual boundary conditions of the membrane in the theoretical model. If fixation by the double-sided tape and jig is insufficient, it is possible that the displacement and bending moment would be zero, corresponding to a simply supported beam.

**Figure 9 micromachines-11-00644-f009:**
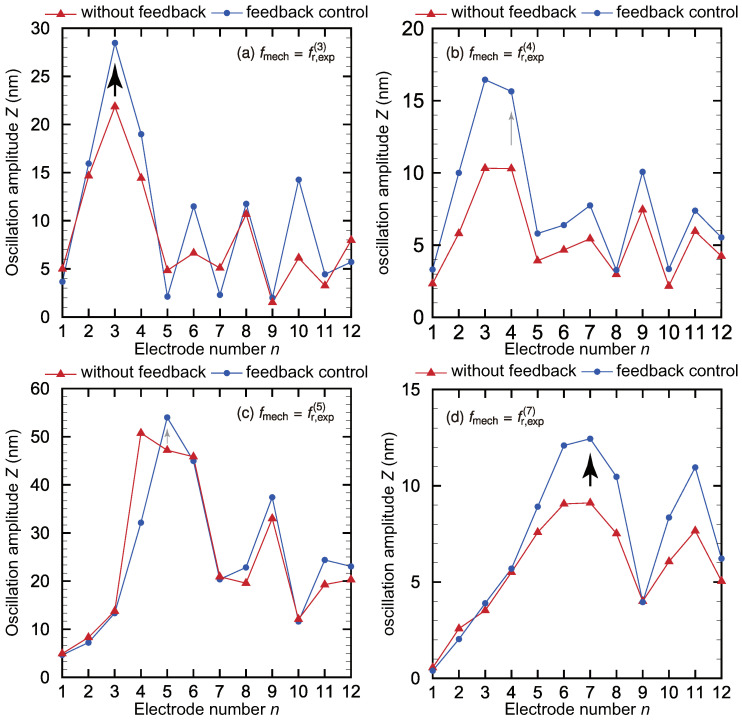
Oscillation amplitudes with and without feedback control at the resonance frequencies corresponding to electrodes (**a**) 3, (**b**) 4, (**c**) 5, and (**d**) 7. The feedback control amplified the oscillation amplitude at the resonance position for all electrodes except for electrode 4, and the frequency selectivity was improved, as quantitatively demonstrated by $Q_{z}$ values in Table 3. Although electrical stimuli were applied to electrodes 3, 4, and 5 to control these oscillations, unexpected behavior occurred at electrode 3, and electrode 4 did not show the maximum oscillation amplitude at its resonance frequency.

**Table 1 micromachines-11-00644-t001:** Fabrication conditions for patterned electrodes.

Process	Solution	Conditions
Spin coating	AZ5214-E	500 rpm, 5 s
		slope, 5 s
		3000 rpm, 30 s
Prebake	–	70${}^{\circ }$, 15 min
UV light exposure	–	6 $\mathrm{mJ}/{\mathrm{cm}}^{2}$
Development and etching	AZ300MIF	approx. 8 min
Rinse	ethanol	2 min

**Table 2 micromachines-11-00644-t002:** Resonance frequencies for each electrode. Based on the linear beam vibration theory, the trapezoidal membrane was modeled as a set of beams extended in *y*-direction, as shown in Figure 1e.

Electrode	1	3	4	5	7	10
Length l(x) [mm]	4.24	4.88	5.20	5.52	6.16	7.12
Resonant frequency $f_{r,\exp}^{\left(n\right)}$ [kHz]	10.5	7.80	6.40	5.40	4.30	3.50
$f_{r,\exp}^{\left(n\right)}l^{2}\left(x\right)$ [${\mathrm{mm}}^{2}\;\mathrm{kHz}$]	189	186	173	165	167	163

**Table 3 micromachines-11-00644-t003:** Evaluation index $Q_{z}$ for the frequency selectivity of the artificial cochlear sensory epithelium. The frequency selectivity was improved for electrodes 3, 5, and 7 by applying local electrical stimuli.

Electrode	3	4	5	7
$Q_{z}$ without feedback control	1.51	1.45	0.978	1.09
$Q_{z}$ with feedback control	1.63	1.41	1.40	1.10

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
