# Peer review of "A Preliminary Prototype High-Speed Feedback Control of an Artificial Cochlear Sensory Epithelium Mimicking Function of Outer Hair Cells"

_micromachines, 2020, doi:10.3390/mi11070644_

Round 1

Reviewer 1 Report

The author propose a novel feedback control technique for the local oscillation amplitude in an artificial cochlear sensory epithelium. The author conducts proper experiments to demonstrate own idea and present the result well. However, there is one thing the author need to be improved in the manuscript. 

In the introduction, the author should start with the background of the topic. Even though the author dealt with it in previous paper, the author should consider the reader who read first from this paper. Then, develop the story and the introduction of the author`s work. It does not seem natural to start by inducing the author`s work.

Author Response

We wish to thank the reviewer for his/her appreciable effort in reviewing the manuscript and for the valuable suggestions. We summarize the reviewer’s comments and our responses in the attachment.

Reviewer 2 Report

This paper presents a piezoelectric PVDF membrane devices and the feedback control algorithm that mimics active control process of the basilar membrane inside the cochlea. The paper demonstrates that the device with feedback control has increased frequency-location selectivity at some electrodes. This paper adds value to the studies of artificial basilar membrane and cochlear implants. I have some comments and questions that are hopefully helpful.

  1. Is there a reason that the two PVDF layer thicknesses are not the same? Usually the voltage output is the maximum for a bimorph when the two layers have the same thickness.
  2. Are the thicknesses of the Dotite conductive adhesive and the CA-191 epoxy measured? It may not be negligible and will affect the device performance.
  3. In line 279, how does the fitted C value compare with the analytical value when calculating from the E, rho, h, l, from equation (1). I do not doubt this analytical equation, however, using the 6 measured data point (with another 4 omitted) to feed the equation to calculate the constant C is not exactly validating the equation prediction, as discussed in line 281.
  4. In figure 6, it might show a better trend and correlation if the displacement and voltage are plotted in log scale.
  5. The stimuli by the actuator and base plate will introduce both the acoustic signal (pressure) and vibration signal (acceleration). It is not super clear which one is the main contributing factor, or both. Or it doesn’t matter in this case. The actuator is optimized to produce constant acceleration in 3-9kHz. Is the acoustic signal that the PVDF membrane sees also flat in 3-9kHz?
  6. It is not clear that if an amplifier or a buffer was used to sense the voltage and if a gain is introduced to the feedback control.

Author Response

(The authors gave the same response as above.)

Reviewer 3 Report

Reviewer’s Recommendation

This candidate paper needs in my opinion major revisions.

Summary

This candidate paper presents a new feedback technique relevant to the local oscillation amplitude in an artificial cochlear sensory epithelium.

General comments

In this candidate paper a lot of work is presented and analyzed. Nevertheless, apart from the innovative facts that are presented, some points inside this presentation should be well-analyzed. Furthermore, I have concluded that in some points this work is not viable, so the authors should try a bit more to emphasize or/and correct some parts of this manuscript. There are reported in the following section. Also, solutions should be given from the authors in some points as their work gives a sense to a point that some systematic mistakes have been done. E.g. The emerged problems with electrodes i.e. data could not be evaluated from some electrodes etc.

Suggested Improvements

Apart from the aforementioned please try to apply the following suggestions and corrections:

  1. Relevant to abstract, please recheck expression and revise accordingly.
  2. You mention that "The repeated high-frequency oscillations 292 cause damage to the fixed end, and the results are not reproducible since the experiments were 293 performed for long periods". A main problem that I understand/foresee is that this system is not yet achieving high levels of fidelity and therefore it would be not viable as a solution to a patient suffering even from hear loss. It would be better to propose a viable future solution to this defect. Consequently, please elaborate.
  3. You mention that various electrodes did not contribute. You must try and report a more in depth scientific explanation.
  4. In figure 5 you should give us the most important emerged fact instead of just reporting the peaks, etc. The previous can be seen and recognized by everyone.
  5. You should include a photo of how your system will be implemented in a human's body and if there is an innovation of its dimensions and its position.
  6. Again in Figure 6, please give us the apothegm.
  7. Finally, my main concerns about this candidate paper are: 1) The measurements as the authors declare have been accomplished not in an environment very close to a real one, 2) Some response times of the system are still too large and they need explanations (e.g. the feedback control procedure takes about 50s); consequently, some restrictions should be mentioned while the title of the candidate paper should change to "A preliminary prototype High-speed feedback control of an artificial cochlear sensory epithelium mimicking function of outer hair cells".

Author Response

(The authors gave the same response as above.)
